# Visualization and Grading of Vitreous Floaters Using Dynamic Ultra-Widefield Infrared Confocal Scanning Laser Ophthalmoscopy: A Pilot Study

**DOI:** 10.3390/jcm11195502

**Published:** 2022-09-20

**Authors:** Gerardo Garcia-Aguirre, Andree Henaine-Berra, Guillermo Salcedo-Villanueva

**Affiliations:** 1Retina Department, Asociación para Evitar la Ceguera en México I.A.P., Mexico City 04030, Mexico; 2School of Medicine and Health Sciences, Tecnologico de Monterrey, Mexico City 14380, Mexico; 3Hospital San Angel Inn, Mexico City 03330, Mexico

**Keywords:** vitreous, vitreous floaters, retina, myodesopsia, scanning laser ophthalmoscopy

## Abstract

Purpose: To describe the appearance of vitreous opacities using dynamic ultra-widefield infrared confocal scanning laser ophthalmoscopy (IRcSLO). Design: Retrospective case series. Methods: Eyes of patients complaining of myodesopsia were analyzed using dynamic ultra-widefield IRcSLO imaging (Nidek Mirante, Nidek Co., Ltd., Gamagori, Japan), and classified according to a vitreous opacity severity scale. Results: Thirty eyes of 21 patients were included in this study. The average age was 56 years. Symptom duration ranged from 1 to more than 365 days. The most common cause of vitreous floaters was posterior vitreous detachment (63.3%), followed by vitreous syneresis (23.3%), asteroid hyalosis (10%) and vitreous hemorrhage (3.3%). Opacities were classified as Grade 1 in three eyes (10%), Grade 2 in 10 eyes (33.3%), Grade 3 in 11 eyes (36.6%), Grade 4 in two eyes (6.6%) and Grade 5 in four eyes (13.3%). Patients with Grade 1 opacities were younger than patients with opacities Grade 2 or greater. A visible Weiss ring could be identified in 0% of eyes with Grade 1 opacities, 40% of eyes with Grade 2 opacities, 100% of eyes with Grade 3 opacities, and 100% of eyes with Grade 4 opacities. In patients with Grade 5 opacities, a Weiss ring could not be identified. Conclusion: Dynamic ultra-widefield IRcSLO imaging is a useful tool to evaluate patients with vitreous floaters. It allows for accurate visualization of the number, density, and behavior of the shadows that vitreous opacities project over a very wide area of the retina, which has a positive correlation with patient perception of floaters.

## 1. Introduction

Vitreous floaters are one of the most frequently encountered problems in clinical practice. It was estimated that they are quite prevalent (76% in a survey of healthy and relatively young smartphone users) [1], and oftentimes cause significant visual disturbance (33% in the same survey).

In spite of their frequency, adequate imaging and grading of vitreous opacities remains elusive, mainly because most opacities that arise from vitreous syneresis or posterior vitreous detachment (PVD), the two most common causes of vitreous floaters [2,3], (1) are semi-transparent, and therefore difficult to resolve from the background during a clinical examination or using fundus photographs, and (2) may be too anterior or too posterior within the vitreous cavity to be adequately imaged with a single device.

Several attempts were made to image and/or quantify vitreous opacities [4], including ultrasonogram [5,6,7,8], time- and spectral-domain optic coherence tomography (OCT) [9,10,11], scanning laser ophthalmoscopy (SLO) [4,10,12,13], and dynamic light scattering [4,6], with varying degrees of success. Among these, infrared confocal SLO (IRcSLO) fundus images have stood out because vitreous floaters, regardless of how posterior or anterior they are, project shadows on the fundus that are clearly visible in grayscale, high-contrast images. 

Another advantage of IRcSLO images is their widespread availability since several devices feature them either as a proper imaging modality or as a preview to obtain other imaging modalities, such as OCT. Most of the former devices, however, use IRcSLO only to capture static images that are averaged. Since the device uses a tracker that adjusts to eye movement, retinal structures are imaged in great detail but vitreous opacities, on the other hand, are mobile even with micro-saccades and when the images are averaged, they are usually seen as blurred and undefined. When IRcSLO images are seen not as a static averaged image but as a video (dynamic), they better reveal the number of floaters, their motility, and the intensity and localization of the shadow that they cast over the retina.

The purpose of this pilot study was to describe the characteristics of vitreous floaters imaged with dynamic ultra-widefield IRcSLO and to classify them according to density and involvement of the posterior pole.

## 2. Materials and Methods

The protocol followed guidelines outlined in the Declaration of Helsinki and was approved by the institutional ethics committee (Comité de Ética en Investigación de la Asociación para Evitar la Ceguera en México, I.A.P., approval number RE-21-24). The study consisted of a retrospective review of charts and ultra-widefield IRcSLO videos of patients whose chief complaint was myodesopsia, that sought care in a private retina practice in Mexico City. Patients with a history of pars plana vitrectomy or significant media opacities were excluded. Patients underwent a complete ophthalmologic examination that included measurement of best corrected visual acuity, intraocular pressure, slit-lamp examination, and dilated fundus evaluation. Degree of symptoms (mild, moderate, severe) as referred by the patient during the initial interview and comorbidities were also recorded. Afterward, structural OCT of the macula and ultra-widefield IRcSLO dynamic images of the affected eye were obtained using the Nidek Mirante (Nidek Co., Ltd., Gamagori, Japan), outfitted with the ultra-widefield (167°) lens adaptor. Since the imaging software interface (NAVIS EX Extra, version 1.11.0.6) did not offer an IRcSLO video capture feature, a video of the computer screen was recorded using a cellphone camera (iPhone 12pro, Apple, Cupertino, CA, USA) in the first 8 cases. After consulting with Nidek advisors, a screen recording software (iFun Screen Recorder, version 1.2.0.261, IOBit, San Francisco, CA, USA) was installed on the computer and used to obtain direct screen recordings. Once the fundus was adequately centered and in focus, patients were instructed to alternately do an upward saccade and return to fixate on the target at the center, with two seconds between saccades, to adequately image the motion of the opacities. Videos were analyzed by a single reader (GG) that was not masked by patient symptoms. Opacities were graded according to a scale that was devised by the authors for the purpose of this study (Table 1, Figure 1, Figure 2, Figure 3, Figure 4, Figure 5, Figure 6 and Figure 7 and Appendix A), considering the density of the opacities (labeled as “diffuse” when some details of the retinal anatomy were visible through the opacity, and as “dense” when no detail was visible through the opacity), and the involvement of the macular area either in primary gaze or after saccadic movements. In addition to grading the opacities, the presence of a visible Weiss ring in the video and the presence of complete PVD as evidenced by structural OCT were recorded.

Data were recorded into spreadsheets using Numbers 11.1 (Apple Inc., Cupertino, CA, USA). Statistical analysis was performed using SPSS version 23 (IBM, Armonk, New York, NY, USA). For qualitative data, we report descriptive statistics. For quantitative data, a non-parametric correlation (Spearman’s test) was performed. Statistical significance was defined as *p* < 0.05.

## 3. Results

Videos from 30 eyes of 21 patients were analyzed for this study (Table 2). Two-thirds of the patients were male. The average age was 56 years. Symptom duration ranged from 1 to more than 365 days. The most common cause of vitreous floaters was posterior vitreous detachment (19 eyes, 63.3%), followed by vitreous syneresis (7 eyes, 23.3%), asteroid hyalosis (3 eyes, 10%) and vitreous hemorrhage (1 eye, 3.3%). 

Opacities were classified as Grade 1 in three eyes (10%), Grade 2 in 10 eyes (33.3%), Grade 3 in 11 eyes (36.6%), Grade 4 in two eyes (6.6%) and Grade 5 in four eyes (13.3%). Patients with Grade 1 opacities were younger than patients with opacities of Grade 2 or greater. Spearman’s test revealed a positive significant correlation between age and opacities (Correlation coefficient: 0.583; *p* = 0.006). A visible Weiss ring could be identified in 0% of eyes with Grade 1 opacities, 4/10 (40%) eyes with Grade 2 opacities, 11/11 (100%) eyes with Grade 3 opacities, 2/2 (100%) eyes with Grade 4 opacities. In patients with Grade 5 opacities, a Weiss ring could not be identified. 

Low-grade opacities (Grade 1–2) were caused by vitreous syneresis (30.7%), PVD (30.7%), myopic vitreopathy (23%) or mild asteroid hyalosis (15.3%). High-grade opacities (Grade 3–4) were all caused by PVD with a visible Weiss ring. Causes of Grade 5 opacities included PVD with significant membrane formation (Figure 7 and Appendix A), hemorrhagic PVD (Figure 8 and Appendix A), vitreous hemorrhage associated with an old central retinal vein occlusion (Figure 9 and Appendix A), and significant asteroid hyalosis (Figure 10 and Appendix A). Patients with a higher grade of opacities had a tendency to be more symptomatic (Figure 11). Spearman’s test revealed a positive significant correlation between opacities and symptoms (correlation coefficient: 0.800, *p* < 0.001).

## 4. Discussion

Vitreous floaters are entoptic phenomena that arise from opacities in the vitreous cavity. These opacities may originate from the natural degeneration of the vitreous that comes with age, resulting in variations in the relationships between water molecules, collagen fibrils, hyaluronic acid, and glycoproteins [14] that lead to vitreous liquefaction (synchysis) or collapse (syneresis) and eventual loss of transparency. They may also originate from posterior vitreous detachment with the presence of a Weiss ring or other phenomena, such as a hemorrhage of inflammation. Symptomatic vitreous opacities were defined as “floaters severe enough to cause symptoms for a minimum time period of 3 months, and which cause enough visual disturbance for the patient to explore therapeutic options” [15].

The real incidence and prevalence of vitreous floaters are unknown. In a retrospective review of patients that underwent vitrectomy for vitreous floaters in Sweden, the incidence was estimated to be 3.1/100,000 [16]. However, this study based its calculation on patients that actively sought surgical treatment for their floaters. In contrast, an electronic survey administered to 603 relatively young smartphone users (average age 29.5 ± 10.7 years) found that 76% of participants saw floaters and 33% reported that these floaters caused a significant disturbance in vision [1].

Historically, the impact of vitreous floaters on visual function has been grossly underestimated [17]. Most patients are often told that floaters will eventually disappear, or that they will grow accustomed to seeing them. However, a survey conducted on patients with floaters showed that they were willing to trade off an average of 1.1 years out of every 10 years of their remaining life to get rid of the symptoms associated with floaters, and were also willing to take a 7% risk of blindness and 11% of death to get rid of symptoms associated with floaters [18]. Additionally, several studies have reported significant improvement in symptoms [16], patient satisfaction [17], quality of life [19,20] and reading speed [20] after vitrectomy for the removal of vitreous floaters.

One of the main factors involved in the underappreciation of the clinical impact of vitreous floaters on visual function is that visual acuity measured using Snellen charts, which is the standard test of visual function in a routine clinical setting, is unable to measure visual disability related with vitreous floaters on everyday functioning and overall quality of life. Another very important factor is that adequate evaluation and imaging of vitreous floaters in vivo remains elusive, since visualizing the vitreous is an attempt to visualize a structure that evolved to be virtually invisible [6], and several diagnostic tools have been used throughout time to evaluate the vitreous humor and its opacities, including ultrasound, OCT and SLO.

Ultrasound was the first method used to systematically evaluate vitreous floaters showing increased prevalence with age using static [5] and dynamic [21] analysis. With this technique, the degree of vitreous opacities was shown to positively correlate with the degradation in contrast sensitivity as well as the patient dissatisfaction index quantified by VFQ measures [22]. Other studies have also validated ultrasound as a reliable tool to grade vitreous floaters [8,23,24]. 

Although ultrasound, especially quantitative (which is not readily available in conventional ophthalmic ultrasound devices), has proven to be a reliable measure of vitreous opacities, there are several areas where it falls short. First, ultrasound only displays one B-scan at a time, and since vitreous opacities are mobile within the vitreous cavity in three dimensions, they become difficult to track accurately. Additionally, the number of opacities visible at any given time depends greatly on the gain settings, unless a standardized value is used, and therefore prone to under- or over-estimation. Furthermore, quantitative ultrasound measurements are performed with the eye in a fixed gaze, and therefore the effect of vitreous opacities moving with head-turning or ocular saccades may be missed. Finally, even though vitreous opacities may be clearly seen in ultrasound, their actual effect (e.g., the shadow that is projected over the retina, and the area of the retina that is affected) is uncertain.

Another imaging modality that was evaluated to visualize vitreous opacities was OCT. This has resulted quite useful in the evaluation of the vitreoretinal interface and cortical vitreous [25] and has a very high agreement with ultrasound for the detection of PVD [26]. Even though the field observable at any given time by OCT used to be narrow (around 6 mm), photo montages were able to show a wider field of the retina and the overlying cortical vitreous [27], and newer devices allow for a larger field (greater than 15mm) [28], or to evaluate the anterior vitreous [11]. Sometimes, if vitreous opacities are significant, they will prevent the scanning laser to reach the retina and will therefore appear as a characteristic shadowing artifact [9].

Although OCT is the imaging method of choice to visualize the vitreoretinal interface and the cortical vitreous when it is attached to the retina, vitreous opacities themselves will only be visible if they are posterior enough to appear in the frame of the OCT, in close proximity to the retina, or if the device is focused anteriorly, where the opacity is freely floating in the vitreous cavity, and therefore, conventional commercially available retinal OCT devices are a suboptimal method to image the central or anterior vitreous. 

Scanning laser ophthalmoscopy was used to visualize intraocular structures for several decades [29,30]. Initially, it was employed as an instrument to image the retina using less energy than indirect ophthalmoscopy or fundus photography [12]. It subsequently underwent several improvements that led to its widespread utilization [31,32], first as a diagnostic tool for glaucoma (that later evolved to become the Heidelberg Retina Tomograph) [33], and eventually gaining ground in the multimodal imaging arena, due to its ability to use different laser wavelengths and to obtain images of outstanding quality. Currently, SLO is employed to obtain fundus images that are focused on different depths according to the wavelength used (infrared for deeper structures and blue or green for more superficial structures), as well as autofluorescence, fluorescein and indocyanine green angiography, and is also widely used for eye tracking and as a reference image (fundus preview) for OCT applications [12].

In the pre-OCT era, SLO had been used to visualize structures in the posterior vitreous, being able to clearly show Weiss rings and other vitreous opacities in static images [13]. However, since obtaining these static, high-definition images of the fundus requires capturing and averaging several “takes” that are fixated on the retina via an eye-tracker, and vitreous opacities are mobile even with micro-saccades, they tend to become blurred and undefined [10]. On the other hand, using dynamic visualization (video), vitreous opacities are highlighted because no image averaging takes place, although some resolution is sacrificed. The images of floaters that are obtained by dynamic IRcSLO are not of the floaters themselves but of the shadow that they cast over the retina, a phenomenon that was compared to “infrared maps of clouds from outer space on the U.S. National Weather Service forecasts” [34].

Several attempts were made to visualize, quantify and/or classify vitreous floaters using IRcSLO. In a study by Garcia et al. [35], the Optos OCT-SLO was used to visualize floaters in patients before and after they developed a PVD. They evaluated contrast sensitivity in these eyes and noted a deterioration when PVD occurred, and an improvement after vitrectomy for floaters. Another study by Vandorselaer et al. [36] used an unspecified SLO device to evaluate vitreous floaters in patients before and after Nd:YAG vitreolysis. Shaimova et al. [37] also evaluated vitreous opacities before and after Nd:YAG vitreolysis using several diagnostic techniques, including structural OCT, OCT angiography with the RTVue xR Avanti, SLO ultra-widefield photographs and B-scan ultrasound. They were able to show the disappearance of large floaters after the procedure. They were also able to quantify the area of the shadow projected on the retina by the floaters using OCT angiography, measuring the area of the shadow with the tool originally intended to measure capillary non-perfusion. Finally, Sun et al. [38] used the Heidelberg Spectralis High-Resolution Angiograph 2 in IRcSLO mode with the 30° or the 55° lens to visualize vitreous floaters. They obtained static, averaged IRcSLO images before and after Nd:YAG vitreolysis, and the opacity of floaters was analyzed using specialized software (ImageJ, version 1.43u, National Institutes of Health, Bethesda, MD, USA). Reduced floater area and better VFQ scores were observed after the procedure.

In our study, we analyzed vitreous floaters using dynamic ultra-widefield IRcSLO imaging, which has not been previously reported. The device used (Nidek Mirante) allows for an ultra-widefield (167°) image of the fundus that shows a more complete picture of the number, density, and movement of vitreous floaters, regardless of etiology. When the vitreous opacity severity scale was designed, we tried to build a grading system that would reflect the symptoms perceived by the patient. That is the reason why the main factors considered were the density of the floaters and the involvement of the posterior pole in primary gaze or after eye movement, which are the most common complaints of patients with floaters. In the patient population studied, there was a positive correlation between higher vitreous opacity grade and patient symptoms.

Even though our scale does not quantify vitreous floaters per se, and even though this technique does not provide their precise location within the vitreous cavity, it allows for clearly visualizing the shadows that vitreous opacities (whether located anteriorly or posteriorly) cast over the retina, both in primary gaze and after eye movement, which could better reflect what the patients are perceiving. When using this technique, it is essential that vitreous opacities are evaluated using dynamic images (video) and not static ones, since in the latter, extraocular (e.g., smudges in the camera lens) or intraocular (e.g., lens, intraocular lens or posterior capsule) opacities may be mistaken for vitreous opacities and affect grading. 

Potential clinical applications for this diagnostic technique include education for patients and their families, reliable documentation of floaters pre- and post-vitrectomy, and ideally, the development of a scale that facilitates decision-making for patients seeking surgical treatment. The videos obtained with this technique may help patients and their families to better understand symptoms and may also guide the clinician to better therapeutic decisions.

Our study has several limitations. First, it was retrospective in nature. In addition, most patients were only asked to do upward saccades, and this may have caused some patients to be classified as Grade 2, when doing saccades in other directions may have classified them as Grade 3. Additionally, the investigator that evaluated the videos was not masked to patient data. Furthermore, videos were obtained using two different capture methods (cellphone camera vs. screen recording software). Since we only recorded the initial patients only with the iPhone, and subsequent patients with the screen capturing software, no comparison was made between both methods. Appendix A were obtained with the screen recording software and the rest with the cellphone camera. According to our observations, both methods of capture allowed us to equally discern between diffuse and dense opacities and to classify them according to our scale, but we acknowledge that the ideal method would be a direct capture from the screen. Lastly, symptoms were only reported subjectively by the patients as mild, moderate, or severe, so the significance of the correlation we observed should be taken with caution. We believe that using a standardized instrument (such as the NEI VFQ) would have been more appropriate. 

In conclusion, it is our impression that dynamic ultra-widefield IRcSLO imaging is a very useful tool to evaluate patients with vitreous floaters. It allows for accurate visualization of the number, density, and behavior of the shadows that vitreous opacities project over a very wide area of the retina, which has a positive correlation with patient perception of floaters. Watching the videos obtained by this technique may help patients and their families to better understand their symptoms and may serve clinicians as a diagnostic tool to decide if a surgical intervention is needed. We acknowledge that this retrospective study has several limitations and therefore we designed a prospective study that has already been approved in our institution to validate this scale using masked observers, NEI VFQ questionnaires and measuring contrast sensitivity to further explore this method of visualizing vitreous opacities and to validate the proposed scale.

## Figures and Tables

**Figure 1 jcm-11-05502-f001:**
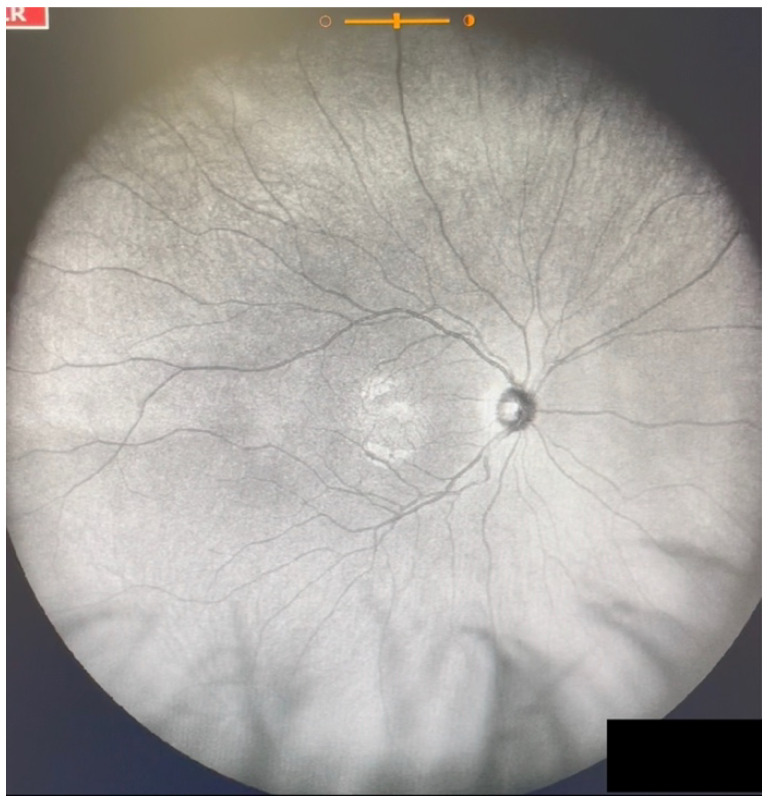
Ultra-widefield IRcSLO image of an eye with Grade 0 vitreous floaters. No shadows are observed. See also Appendix A.

**Figure 2 jcm-11-05502-f002:**
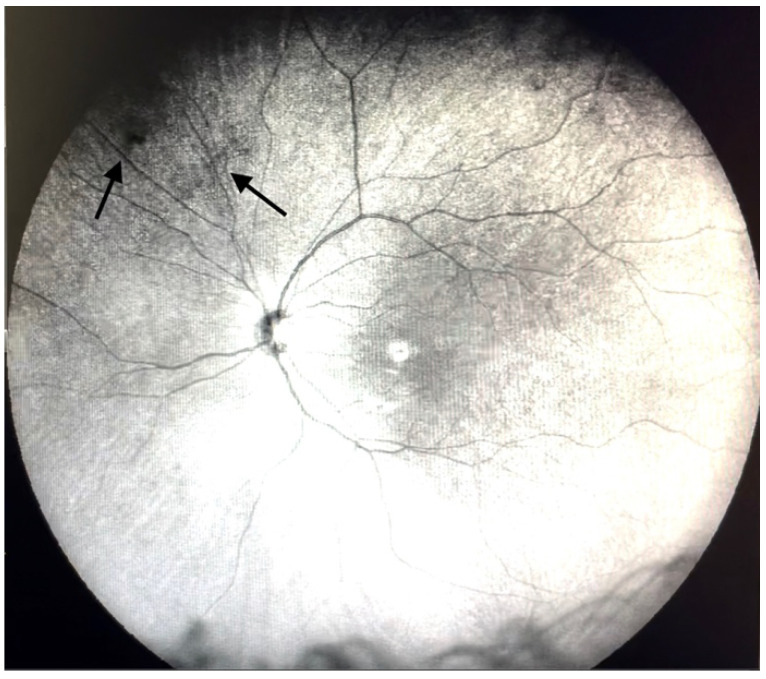
Ultra-widefield IRcSLO image of an eye with Grade 1 vitreous floaters. Some diffuse shadows are observed in the superonasal periphery (arrows). See also Appendix A.

**Figure 3 jcm-11-05502-f003:**
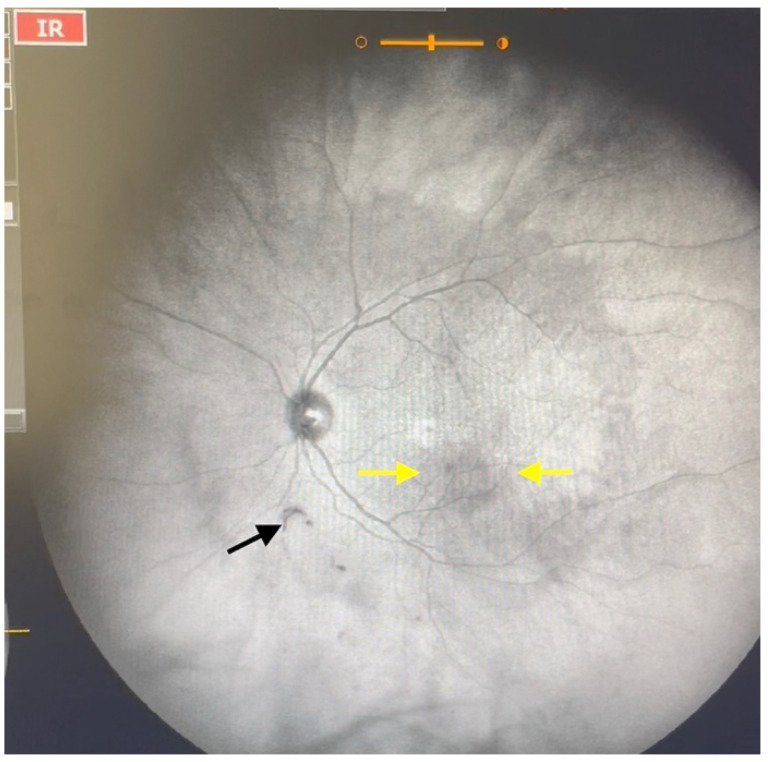
Ultra-widefield IRcSLO image of an eye with Grade 2 vitreous floaters. Some diffuse shadows are observed within the macular area (yellow arrows). A Weiss ring is also visible outside the macular area (black arrow). See also Appendix A.

**Figure 4 jcm-11-05502-f004:**
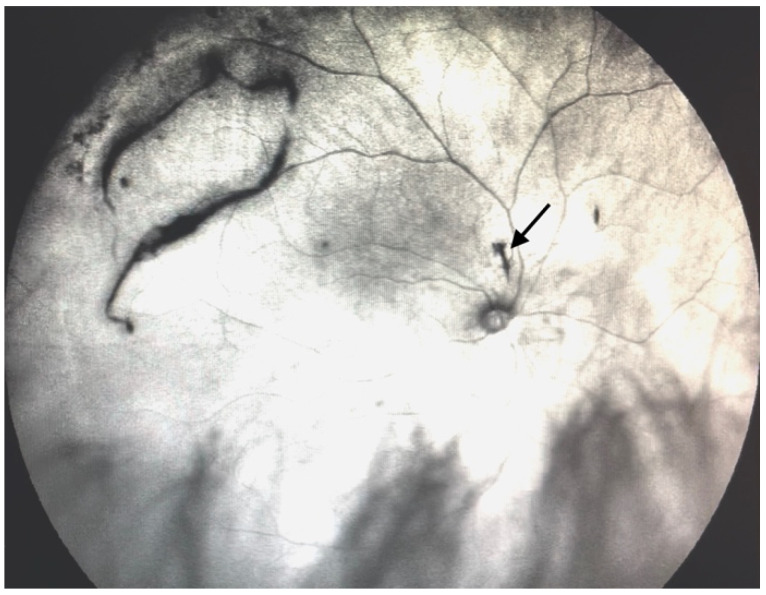
Ultra-widefield IRcSLO image of an eye with Grade 2 vitreous floaters. Dense shadows are observed outside the macular area, in the superotemporal periphery, overlying an area of lattice degeneration. A Weiss ring is also visible outside the macular area (arrow). See also Appendix A.

**Figure 5 jcm-11-05502-f005:**
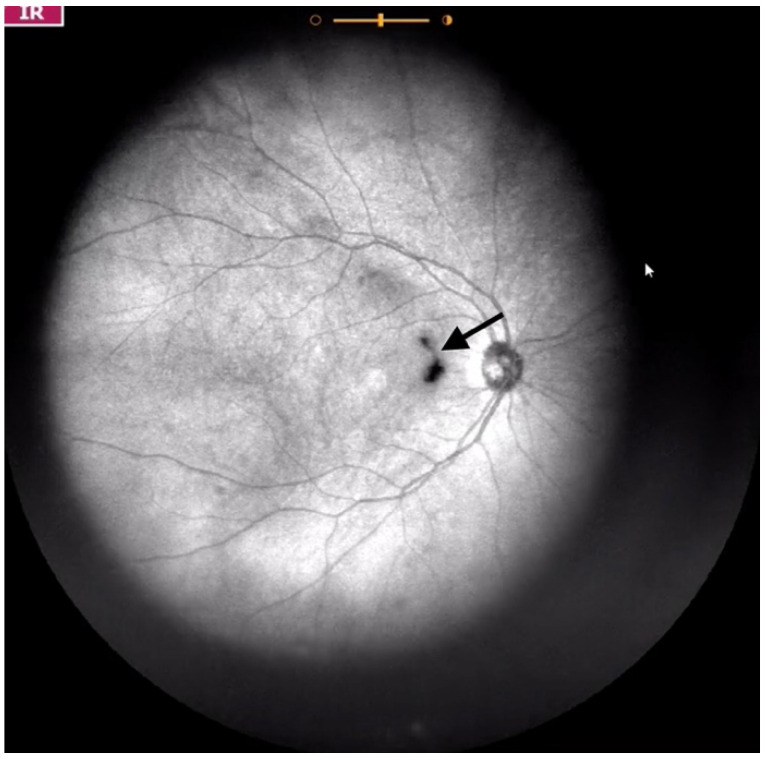
Ultra-widefield IRcSLO image of an eye with Grade 3 vitreous floaters. This still image was obtained after an upward saccade and shows a dense shadow inside the macular area (arrow). See also Appendix A.

**Figure 6 jcm-11-05502-f006:**
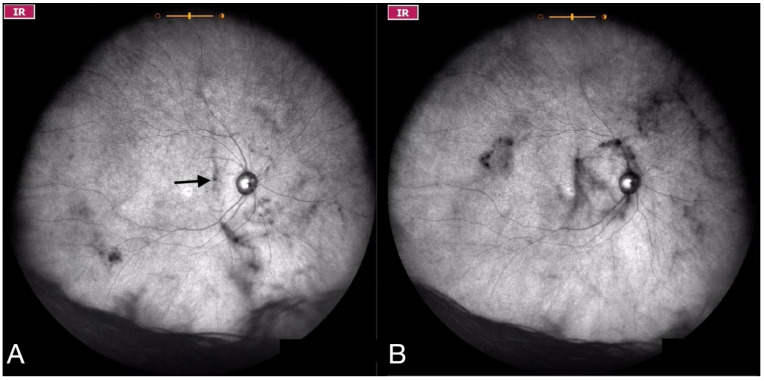
Ultra-widefield IRcSLO images of an eye with Grade 4 vitreous floaters. (**A**) Several diffuse shadows are observed when the eye is in primary gaze. A dense shadow is observed in the macular area (arrow). (**B**) Vitreous opacities are stirred after a saccadic movement and denser shadows are observed inside the macular area. See also Appendix A.

**Figure 7 jcm-11-05502-f007:**
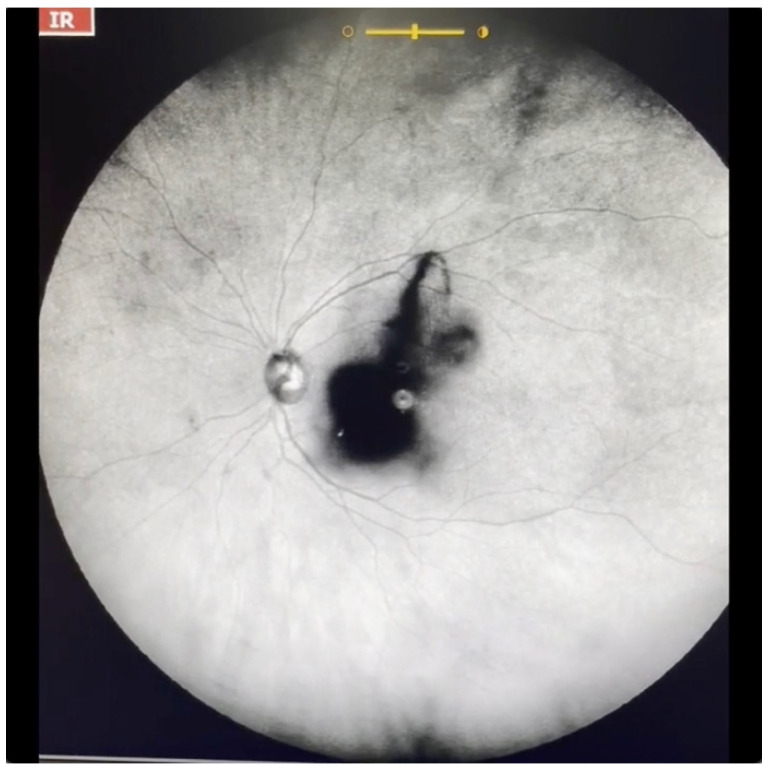
Ultra-widefield IRcSLO image of an eye with Grade 5 vitreous floaters secondary to posterior vitreous detachment. A very dense shadow is observed inside the macular area. See also Appendix A.

**Figure 8 jcm-11-05502-f008:**
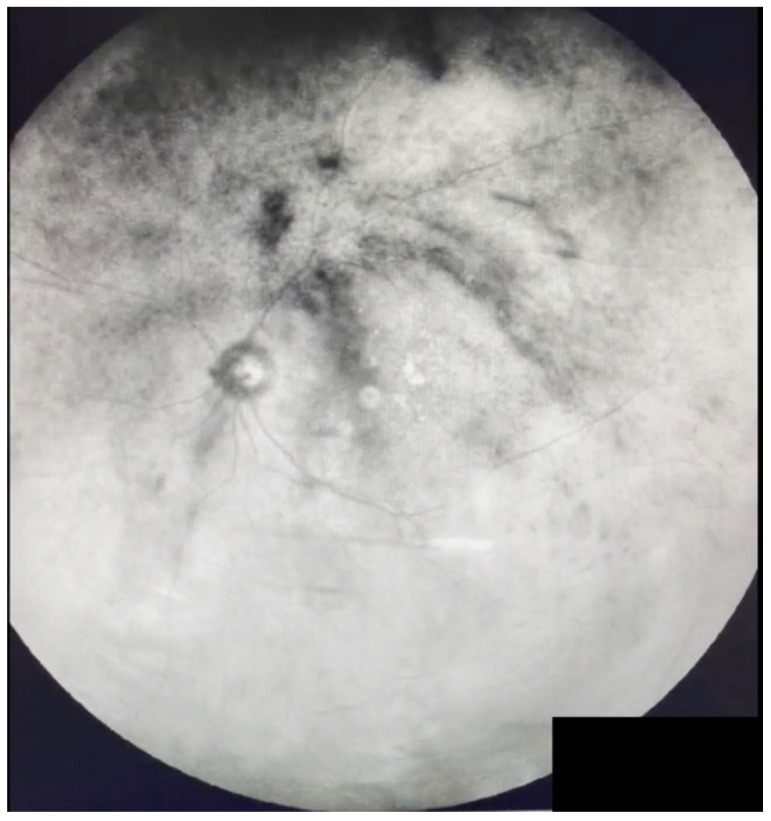
Ultra-widefield IRcSLO image of an eye with Grade 5 vitreous floaters secondary to hemorrhagic posterior vitreous detachment. Dense shadows are observed within the macular area in the primary gaze. See also Appendix A.

**Figure 9 jcm-11-05502-f009:**
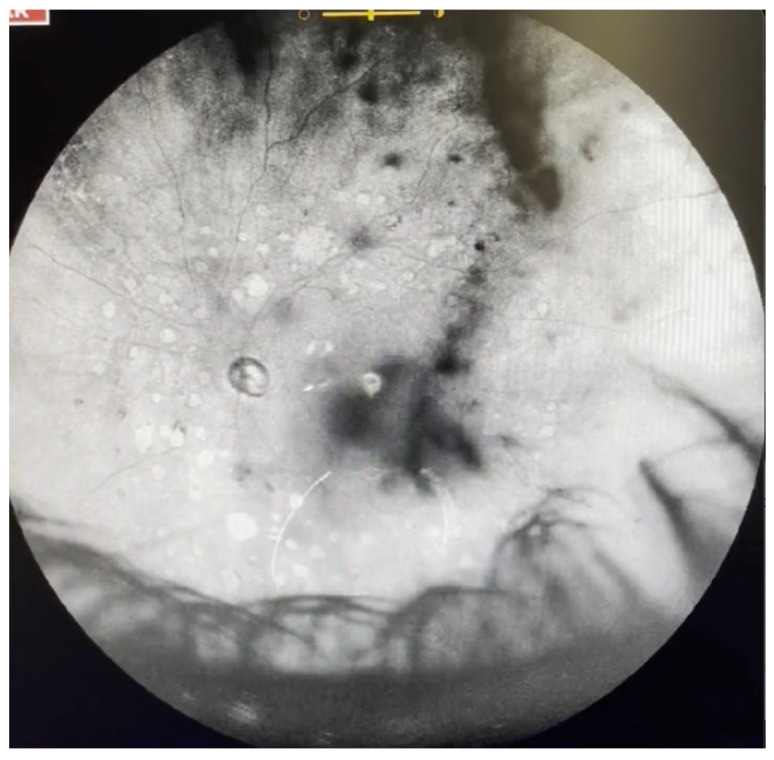
Ultra-widefield IRcSLO image of an eye with Grade 5 vitreous floaters secondary to vitreous hemorrhage in a patient with history of central retinal vein occlusion that was treated with laser photocoagulation. Dense shadows are observed within the macular area in the primary gaze. See also Appendix A.

**Figure 10 jcm-11-05502-f010:**
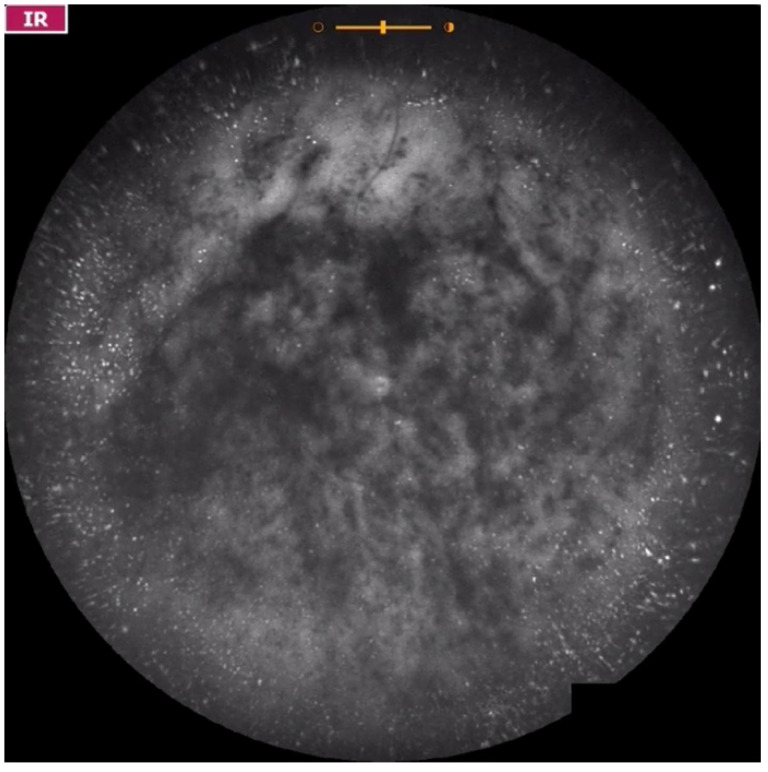
Ultra-widefield IRcSLO image of an eye with Grade 5 vitreous floaters secondary to vitreous hemorrhage in a patient with history of central retinal vein occlusion that was treated with laser photocoagulation. Dense shadows are observed within the macular area in the primary gaze. See also Appendix A.

**Figure 11 jcm-11-05502-f011:**
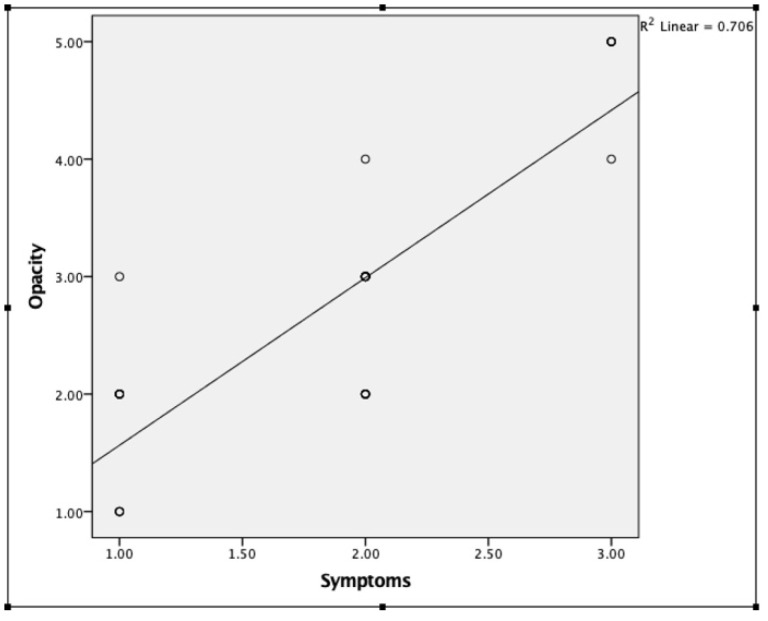
Spearman’s test revealed a positive significant correlation between opacities and symptoms (correlation coefficient: 0.800, *p* < 0.001).

**Table 1 jcm-11-05502-t001:** Vitreous opacities severity scale.

Grade 0—No vitreous opacities visible (Figure 1 and Appendix A).
Grade 1—Diffuse opacities that do not cross the center of the macula (Figure 2 and Appendix A).
Grade 2—Diffuse opacities that cross the center of the macula (Figure 3 and Appendix A) and/or dense opacities that do not cross the center of the macula (Figure 4 and Appendix A).
Grade 3—Dense opacities outside the center of the macula in primary gaze, that cross the center of the macula with eye movement (Figure 5 and Appendix A).
Grade 4—Dense opacities that involve the center of the macula in the primary gaze (Figure 6 and Appendix A).
Grade 5—Dense opacities that obstruct at least 30% of the macula (~2 disc diameters in size) (Figure 7 and Appendix A).

**Table 2 jcm-11-05502-t002:** Patient data.

Patient Number	Gender	Age (Years)	Eye	Time from Beginning of Symptoms (Days)	Severity of Symptoms	Diagnosis	Visible Weiss Ring	PVD per Structural OCT	Vitreous Opacities Severity Scale	Comorbidities
1	M	39	OD	180	+	Myopic vitreopathy	NO	NO	2	High myopia
			OS	180	+	Myopic vitreopathy	NO	NO	1	High myopia
2	F	25	OD	365	+	Syneresis	NO	NO	1	
3	F	24	OS	10	+	Syneresis	NO	NO	1	
4	M	70	OD	>365	+	Asteroid hyalosis	NO	NO	2	Asteroid hyalosis
			OS	1	++	PVD	YES	YES	2	
5	F	32	OS	60	++	Myopic vitreopathy	NO	NO	2	High myopia
6	M	59	OD	>365	+	Asteroid hyalosis	NO	NO	2	Asteroid hyalosis
			OS	>365	+++	Asteroid hyalosis	NO	N/A	5	Asteroid hyalosis
7	M	62	OD	>365	++	PVD	YES	YES	3	
			OS	>365	++	PVD	YES	YES	2	
8	F	59	OD	>365	++	PVD	YES	YES	2	Peripheral lattice degeneration
9	M	61	OS	30	++	PVD	YES	YES	2	
10	M	55	OD	120	++	PVD	YES	YES	3	
			OS	120	+	Syneresis	NO	NO	2	
11	M	60	OD	180	+	Syneresis	NO	NO	2	
			OS	180	++	PVD	YES	YES	3	
12	M	64	OS	4	++	PVD	YES	YES	3	Diabetic macular edema
13	M	64	OD	90	+	PVD	YES	YES	3	Cataract
			OS	21	++	PVD	YES	YES	3	Cataract
14	M	47	OD	11	++	PVD	YES	YES	3	
15	M	66	OD	330	++	PVD	YES	YES	3	
16	M	56	OD	7	+++	PVD	YES	YES	4	
			OS	90	++	PVD	YES	YES	3	
17	F	69	OD	120	++	PVD	YES	YES	3	
			OS	60	++	PVD	YES	YES	3	
18	M	58	OS	270	++	PVD	YES	YES	4	
19	F	62	OS	10	+++	Hemorrhagic PVD	NO	YES	5	
20	M	81	OS	5	+++	Hemorrhagic PVD	NO	YES	5	Central retinal vein occlusion
21	M	63	OS	>365	+++	PVD	NO	YES	5	

Abbreviations: M—Male, F—Female, OD—Right eye, OS—Left eye, +—Mild, ++—Moderate, +++—Severe, PVD—Posterior vitreous detachment, OCT—Optic coherence tomography.

## Data Availability

Data are available upon request at jerry_gar@me.com.

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
