# Peer review of "Visualization and Grading of Vitreous Floaters Using Dynamic Ultra-Widefield Infrared Confocal Scanning Laser Ophthalmoscopy: A Pilot Study"

_jcm, 2022, doi:10.3390/jcm11195502_

Round 1

Reviewer 1 Report

1.     Did the authors exclude eyes with vitirtis?

2.     Eyes having vitreous hemorrhage (VH) might be in different stages, like early VH, resolving VH, etc. This needs to be taken into consideration during the interpretation of results.

3.     How did the authors classify the symptoms of the patients related to myodesopsia as mild moderate and severe? From retrospective data, it will be even more challenging.

4.     The authors mention that Since the imaging software interface (NAVIS EX Extra, version 1.11.0.6) did not offer an IRcSLO video capture feature, a video of the computer screen was recorded using 77 either a cellphone camera (iPhone 12pro, Apple, Cupertino, California) or a screen recording software (iFun Screen Recorder, version 1.2.0.261).

This statement carries a lot of importance regarding the methodology of the study.

1.     Was there a correlation between the cellphone and screen recording software?

2.     Since the video recorded by the cellphone or the screen recording software was used for analysis, this should be mentioned as a limitation. The ideal method would have been capturing video from the imaging software interface.

5.     Opacities were graded according to a scale that was devised by the authors for the purpose of this study. Did the authors do any pilot study to validate this grading?

6.     Grade 5 mentions about 30% of the macula. How to measure this 30%?

7.     The authors need to mention the artifacts, including those in the lens of the fundus camera system, which can cause a shadow and look like vitreous opacities, in this black and white image, though not in the videos.

8.     In the discussion, the authors discuss extensively the existing literature and knowledge on vitreous floaters and their imaging. However, the discussion should focus more on the current study. The redundant part may be cut-short.

9.     The clinical application of this study needs to be discussed.

10.  A prospective study with a greater number of study participants would be better in this field of literature. The authors can add this sentence to the manuscript.

Author Response

Dear Doctor:

Thank you for the time and effort to review our manuscript, and for the insightful comments that will surely improve our paper. We have made the following adjustments to the manuscript:

1.     Did the authors exclude eyes with vitirtis?
We did not purposefully exclude eyes with vitritis, but no eyes with vitritis presented during the study.

2.     Eyes having vitreous hemorrhage (VH) might be in different stages, like early VH, resolving VH, etc. This needs to be taken into consideration during the interpretation of results.
    We believe the stage of vitreous hemorrhage (and actually the etiology of any of the cases of vitreous opacities) is not relevant to how the opacities are observed with this technique, and how they fit in the classification.

3.     How did the authors classify the symptoms of the patients related to myodesopsia as mild moderate and severe? From retrospective data, it will be even more challenging.
    At our center, patients presenting with myodesopsia are routinely asked to subjectively classify the symptoms they are experiencing as mild, moderate or severe. No objective measure of symptoms is performed. A prospective study using this visualization and grading system has been approved by our IRB and is currently underway (IRB approval number RE-22-03), where patients will have to answer a VFQ questionnaire, they will undergo a contrast sensitivity test, and masked readers will classify the opacities by watching the videos. A statement in this regard was added to the discussion (LINE 341)

4.     The authors mention that Since the imaging software interface (NAVIS EX Extra, version 1.11.0.6) did not offer an IRcSLO video capture feature, a video of the computer screen was recorded using either a cellphone camera (iPhone 12pro, Apple, Cupertino, California) or a screen recording software (iFun Screen Recorder, version 1.2.0.261). This statement carries a lot of importance regarding the methodology of the study.
    The reason we used these two recording methods was because at first no screen recording software was installed in the Mirante computer. After seeing how this could be a good method to visualize (and maybe grade) vitreous opacities, we started recording videos with the iPhone and showed them to people in Nidek. We then asked them if we could record the screen directly, and they were the ones who recommended the iFun Screen Recorder, which is what we used to record videos with the rest of the patients. A statement in this regard was added to the methods section (LINES 76 and 324)

5.     Was there a correlation between the cellphone and screen recording software?
    Since we only recorded the initial patients only with the iPhone, and subsequent patients with the screen capturing software, no comparison was made between both methods. Videos 5,6 and 10 were obtained with iFun, and the rest with the iPhone. According to our observations, both methods of capture allowed us to equally discern between diffuse and dense opacities, and to classify them according to our scale. A statement was added to the discussion in this regard (LINE 324).

6.     Since the video recorded by the cellphone or the screen recording software was used for analysis, this should be mentioned as a limitation. The ideal method would have been capturing video from the imaging software interface.
We agree with this, and a statement was added to the discussion (LINE 330)

5.     Opacities were graded according to a scale that was devised by the authors for the purpose of this study. Did the authors do any pilot study to validate this grading?
No pilot study was performed. This manuscript was meant to be the first report of this technique to visualize floaters, and to propose the scale. The fact that this is a pilot study was added to the title (LINE 4). A prospective study has been approved in our institution to validate this scale using masked observers, NEI VFQ questionnaire and measuring contrast sensitivity. A statement was added to the discussion (LINE 341).

6.     Grade 5 mentions about 30% of the macula. How to measure this 30%?
This is roughly 2 disc diameters, and was added to the scale (LINE 109)

7.     The authors need to mention the artifacts, including those in the lens of the fundus camera system, which can cause a shadow and look like vitreous opacities, in this black and white image, though not in the videos.
This is a great point. Artifacts such as the ones mentioned, are easy to distinguish from vitreous opacities in video, since the former, if present, are static, while vitreous opacities are mobile. That is not the case with static images. A statement in this regard has been added to the discussion (LINE 309)

8.     In the discussion, the authors discuss extensively the existing literature and knowledge on vitreous floaters and their imaging. However, the discussion should focus more on the current study. The redundant part may be cut-short. 
The length of that part of the discussion has been reduced significantly (~30%).

9.     The clinical application of this study needs to be discussed. 
A paragraph has been added to the discussion (LINE 314)

10.  A prospective study with a greater number of study participants would be better in this field of literature. The authors can add this sentence to the manuscript.
A prospective study has already been approved. A statement has been added at the end of the discussion (LINE 341).

Reviewer 2 Report

The authors present a study of imaging vitreous opacities on IRSLO in various vitreoretinal conditions. Hence, the clinical application of the various grades of opacities remains unclear. For example, asteroid hyalosis patients present with floaters only when associated with other conditions like a vitreous haemorrhage or acute PVD. A control group of asymptomatic patients could have been useful.

Author Response

Dear Doctor:

Thank you for the time and effort to review our manuscript, and for the insightful comments that will surely improve our paper. We have made the following adjustments to the manuscript:

  1. The authors present a study of imaging vitreous opacities on IRSLO in various vitreoretinal conditions. Hence, the clinical application of the various grades of opacities remains unclear. For example, asteroid hyalosis patients present with floaters only when associated with other conditions like a vitreous haemorrhage or acute PVD. A control group of asymptomatic patients could have been useful. 

We believe that a favorable aspect of this technique is that regardless of the etiology, the shadow of vitreous opacities in the retina is accurately represented, and may better reflect patient symptoms than ultrasound of OCT. A prospective study has been approved and is currently underway in our institution to validate this scale using masked observers, NEI VFQ questionnaires and measuring contrast sensitivity to further explore this method of visualizing vitreous opacities and to validate the proposed scale. A statement in this regard was added in the discussion (LINE 341).

Reviewer 3 Report

Errors in citation pages 1-2, 10

Unmasked grader a significant limitation that makes results essentially useless, as correlation to symptoms is heavily biased. Would reframe this as a pilot project to emphasize describing the technique of capturing the imaging and describe how future research will need to be performed in a blinded fashion, with an increased sample size. Other authors have described the use of more objective measures of visual significance of floaters, such as contrast sensitivity - this would be helpful to include for future discussion rather than subjective grading of symptoms. 

Author Response

Dear Doctor:

Thank you for the time and effort to review our manuscript, and for the insightful comments that will surely improve our paper. We have made the following adjustments to the manuscript:

1. Errors in citation pages 1-2, 10
Errors have been corrected.

2. Unmasked grader a significant limitation that makes results essentially useless, as correlation to symptoms is heavily biased. Would reframe this as a pilot project to emphasize describing the technique of capturing the imaging and describe how future research will need to be performed in a blinded fashion, with an increased sample size. Other authors have described the use of more objective measures of visual significance of floaters, such as contrast sensitivity - this would be helpful to include for future discussion rather than subjective grading of symptoms. 

This manuscript was meant to be the first report of this technique to visualize floaters, and to propose the scale. The fact that this is a pilot study was added to the title (LINE 4). A prospective study has been approved and is currently underway in our institution to validate this scale using masked observers, NEI VFQ questionnaires and measuring contrast sensitivity to further explore this method of visualizing vitreous opacities and to validate the proposed scale. A statement in this regard was added in the discussion (LINE 341).

Round 2

Reviewer 1 Report

1. Thanks for making the corrections.

2. the reference numbers need to be updated, as per the shortening of the text in the main manuscript.

Author Response

Dear reviewer:

1. Thanks again for your comments and evaluation of our manuscript.

2. Corrections were made to the references.

Reviewer 2 Report

The authors have improved the script as suggested

Author Response

Dear reviewer:

Thank you for your comments and for letting us improve our manuscript. Some corrections were made to reference numbering.